# Derailed Ceramide Metabolism in Atopic Dermatitis (AD): A Causal Starting Point for a Personalized (Basic) Therapy

**DOI:** 10.3390/ijms20163967

**Published:** 2019-08-15

**Authors:** Markus Blaess, Hans-Peter Deigner

**Affiliations:** 1Institute of Precision Medicine, Medical and Life Sciences Faculty, Furtwangen University, Jakob-Kienzle-Strasse 17, 78054 Villingen-Schwenningen, Germany; 2EXIM Department, Fraunhofer Institute IZI Leipzig, Schillingallee 68, 18057 Rostock, Germany

**Keywords:** atopic dermatitis, lysosomotropic compounds, apoptosis, lysosome, ceramide metabolism, amitriptyline, ceramide de novo synthesis, antioxidants, linoleic acid, sphingolipid profile

## Abstract

Active rebuilding, stabilizing, and maintaining the lipid barrier of the skin is an encouraging disease management and care concept for dry skin, atopic dermatitis (eczema, neurodermatitis), and psoriasis. For decades, corticosteroids have been the mainstay of topical therapy for atopic dermatitis; however, innovations within the scope of basic therapy are rare. In (extremely) dry, irritated, or inflammatory skin, as well as in lesions, an altered (sphingo)lipid profile is present. Recovery of a balanced (sphingo)lipid profile is a promising target for topical and personalized treatment and prophylaxis. New approaches for adults and small children are still lacking. With an ingenious combination of commonly used active ingredients, it is possible to restore and reinforce the dermal lipid barrier and maintain refractivity. Lysosomes and ceramide de novo synthesis play a key role in attenuation of the dermal lipid barrier. Linoleic acid in combination with amitriptyline in topical medication offers the possibility to relieve patients affected by dry and itchy skin, mild to moderate atopic dermatitis lesions, and eczemas without the commonly occurring serious adverse effects of topical corticosteroids or systemic antibody administration.

## 1. Introduction

Atopic dermatitis (AD), atopic eczema, or neurodermatitis is a chronic or chronic relapsing, non-contagious skin disease. Morphology and localization is age-dependent pronounced differently and frequently accompanied by severe itching [1,2,3]. Consensus-based European guidelines for treatment of AD are usually associated with sometimes serious adverse events that reduce patient compliance or adherence to treatment and increase the risk of bacterial or virus infections [3,4]. Glucocorticoids, which cause changes in skin structure, are limited in duration of use. With atrophy of the skin at the forefront and in the intertrigenes (armpit, groin), there is an additional risk of superinfections and of triggering rosacea, steroid acne in the face, development of telangiectasias, and spontaneous scars [3,4]. Permanent daily treatment is not recommended. Topical calcineurin inhibitors tacrolismus and pimecrolimus cause transient warmth, tingling, or a burning sensation at the application site during the first days of application and increase the risk of bacterial or viral skin infections [3,4].

Basic therapy for disturbed skin barrier function and emollient therapy of AD includes the topical application of active, compound-free ointment bases with an occlusion effect, which is intended to prevent water loss from outer layers of the skin (e.g., through liquid paraffin or ceresin as an occludent) and improvement in retaining moisture (e.g., through urea as humectant) or the addition of extra water into dry skin using hydrophilic creams. Fatty ointments for dry skin or hydrating oil-in-water emulsions on less dry skin are used. However, many patients do not respond in the expected manner, becoming frustrated with the efficacy of the drugs and fearful of adverse effects [5].

The major goal of any advanced disease management concept is to improve therapeutic effectiveness, increase patient satisfaction with treatment, and minimize adverse effects. Symptomatic therapy (inflammation: glucocorticoids; itching: H_1_-receptor antagonists or local anesthetics (polidocanol)) are often unable to restore skin integrity. Thus, a precise knowledge of the role of lysosomes in the pathological derailment of ceramide metabolism in AD enables an advanced disease management concept that may replace some common topical (basic) therapies in the near future. The transition from symptomatic to causal topical treatment will provide patients with long-term relief of their symptoms.

Starting from the altered ceramide profile and the appearance of C_16_-ceramide in the skin of AD patients, we try to point out the underlying causes and to describe how these alterations can be eliminated by topical therapy in order to restore the original complexion. Our modular disease management concept renders humectants and occludents unnecessary. Targeting one of the AD root causes and AD pathogenesis at an early stage, our suggested therapy is well-suited for patients independent of acute or chronic infections. Our therapeutic approach is applicable to all age groups with just fractional adverse effects.

## 2. Ceramides and Altered Sphingolipid Profile in AD

Ceramide is a general term for certain well-characterized sphingolipid metabolites and second messengers involved in numerous biological processes in the cell. They consist of a backbone (dihydrosphingosine (sphinganine) [dS] sphingosine [S], phytosphingosine [P], or 6-hydroxy-sphingosine [H]) and a fatty acid residue (non-hydroxy fatty acid [N], α-hydroxy fatty acid [A], or esterified ω-hydroxy fatty acid [EO]) (Figure 1A). The combination of fatty acid and backbone results in 12 subclasses of ceramide fractions of the stratum corneum.

Cell cycle-relevant C_16_-ceramide and C_24:1_-ceramide (backbone: sphingosine, residue: non-hydroxy fatty acid) belong to subclass NS, representing 7% of total ceramide of the skin (Figure 1B). Subclasses NH (23%), NP (22%), AP (16%), and AH (15%) represent the majority of dermal ceramides. Cellular ceramides typically have fatty acid residues that range from 16 to 26 carbon atoms; in the stratum corneum; ceramides of up to 32 carbon atoms exist [6].

### 2.1. Role of Ceramides in Cells and Skin Development

Ceramides are not only a key molecule in sphingolipid metabolism that can be modified in cells (Figure 2), but are also important signaling molecules that are able to regulate vital cellular functions [7]. As cellular messengers, they are able to halt the cell cycle (G0/G1 and G2/M arrest) and promote apoptosis [8,9,10]. The length of the fatty acid residue is decisive for the function or individual effect of each ceramide. Ceramides, in particular C_16_-ceramide, are strongly increased in apoptotic cells and play a crucial role in apoptosis [11]. On the contrary, (very) long-chain ceramides (>C_20_) are not pro-apoptotic, and are therefore responsible for the epidermal barrier function [12].

In epidermal keratinocytes, ceramide is synthesized and released into the extracellular matrix of the stratum corneum. There, ceramides are an essential element of the extracellular lipid matrix and play an critical role in the formation of the most important natural barrier to water loss and the penetration of various compounds into the skin [12,13]. During keratinocyte maturation, total ceramide increases from 3.8 ± 0.2% (stratum basale/spinosum) to 18.1 ± 0.4% (stratum corneum). Inflammatory skin diseases are accompanied by a shift in the proportion of ceramide subclasses. In AD, the subclass AP is increased; in psoriasis, the subclasses AP, NP, and EOP are lowered [6]. Various care products attempt to compensate alterations in affected areas by local ceramide replacement therapy using ceramide 1 (EOP), ceramide 3 (NP), and ceramide 6 (AP) (Figure 1C).

### 2.2. (Sphingo)Lipid Profile of Sensitive and Inflammatory Skin and Lesions of Patients with AD

Experimental findings in patients with AD display a change in the composition of skin lipids (Table 1). In skin lesions, the content of saturated free fatty acids with very long carbon chains (≥C_24_) is significantly reduced, whereas short-chain free fatty acids—in particular, palmitic acid (C_16:0_) and stearic acid (C_18:0_)—are increased. Analysis of the fatty acid residues of ceramides showed substantially increased amounts of short-chain ceramides (<42 carbon atoms, especially C_16_-ceramide and C_18_-ceramide [NS]; on the contrary, long-chain ceramides (>42 carbon atoms, e.g., C_24:1_-ceramide) were significantly reduced [6,14,15,16]. Significantly altered distribution was observed predominantly in skin lesions and in non-lesional skin areas of patients with AD. Among the short-chain ceramides, C_16_-ceramide (C34-ceramide) in particular is increased. The level of C_16_-ceramide is the highest in the lesions of patients with AD compared to the skin of healthy individuals [14]. An increase in free palmitic acid indicates an increased degradation of lipids containing palmitic acid from cell membranes of keratinocytes and/or release from palmitoyl-CoA of fatty acid biosynthesis. Mouse models of mice that develop AD also show these changes in ceramide profile [13,17].

### 2.3. Impaired Maturation and Premature Apoptosis of Keratinocytes

Apoptosis (controlled cell death) is the most important regulatory element in the control of cell populations, organ formation, histological structures, skin, and the related dermal lipid barrier [18]. In contrast to healthy skin, C_16_-ceramide is elevated in the vulnerable skin areas and lesions of AD patients and in AD mouse models [13,14,17], suggesting that (premature) apoptosis has been induced there. A disorder in the application of this highly specific tool may explain the incomplete maturation of keratinocytes, the defective lipid barrier, and the subsequent inflammatory process. Due to premature apoptosis, a high percentage of keratinocytes in lesions or severely reddened, irritated, or inflammatory skin do not reach full maturity, including the ceramide content required to form a stable cell cluster as a barrier against moisture loss. Figure 3 illustrates how C_16_-ceramide can be selectively produced in impacted cells. Oxidative stress (lack of NAD(P)H and GSH) in cells not only triggers lysosomal C_16_-ceramide formation, but also prevents the synthesis of very long-chain ceramides at the endoplasmic reticulum (Figure 3 and Figure 4).

### 2.4. The Two Different pH-Dependent Enzymatic Activities of aCERase

Interestingly, lysosomal acidic ceramidase (aCERase) exhibits two different enzymatic activities dependent on the lysosomal pH: the well-known ceramide hydrolase activity (optimum pH 4.0–5.0) and a little-known ATP-independent reverse ceramide synthase activity (revaCERase) (optimum pH 5.5–6.5). revaCERase preferentially reacts palmitic acid (C_16_) and stearic acid (C_18_) with sphingosine [19,20]. Unlike ceramide synthases (CerS), no activation of fatty acid via acyl-CoA is required. Sphingomyelin of cell membranes contains a high amount of palmitic acid and only very little stearic acid [21,22]. Therefore, the selectivity of the re-synthesis between C_16_-ceramide and C_18_-ceramide is due to the availability of each fatty acid in the lysosome. The objective of a well-suited therapy is to stabilize the physiological pH value of lysosomes in keratinocytes.

### 2.5. aCERase: Shift from Ceramidase to revaCERase Activity in Lysosomes

Acidic pH (4.6–4.8) in lysosomes is predominantly generated and maintained by the vacuolar H^+^-ATPase (V-ATPase). The lysosomal redox chain is the second proton pump transporting protons into the lumen of lysosomes. V-ATPase uses ATP, the lysosomal redox chain uses NADH from cytoplasm as an energy source [23,24]. The catalytically active subunit (73 kDa) of V-ATPase contains a conserved region (P-LOOP) with two cysteines at positions 254 and 532, which are capable of forming disulfide bonds, thus inactivating the active site of V-ATPase. The thiol–disulfide equilibrium depends on the redox potential of the cytoplasm [23]. An excessive drop of cytosolic redox potential leads to the formation of the disulfide bond between Cys 254 and Cys 532, blocking the catalytically active ATPase site. Once formed, the disulfide bond is quite stable under physiological conditions. Physiological glutathione (GSH) concentrations in the cytoplasm are incapable of reconstituting both cysteines and recovering full V-ATPase activity [25]. A consequence of the failure of V-ATPase is the inversion of CERase activity: ceramide hydrolysis vanishes, while revaCERase activity selectively generates C_16_-ceramide in the lysosome, and triggers the premature apoptosis of keratinocytes (Figure 3 and Figure 4).

C_16_-ceramide is formed of sphingosine and palmitic acid. Besides the hydrolysis of membranous sphingomyelin, the sources of the required palmitic acid are the hydrolysis of accumulating C_16_-CoA (by palmitoyl-CoA hydrolase) and the hydrolysis of phosphatidylcholine (by lysosomal phospholipase A_2_) (Figure 4B).

## 3. New Therapy Concept: Active Adjustment of Ceramide Metabolism

Within these findings, a new therapeutic target for the therapy of AD can be identified: reinforcement of the dermal lipid barrier by the active normalization of (sphingo)lipid metabolism and very long chain acyl-CoA biosynthesis in keratinocytes. As a key molecule in the sphingolipid metabolism, ceramides are likewise a product and starting material of various enzymatic modifications in cells. Target-oriented normalization of the altered ceramide metabolism in keratinocytes is a promising approach for effective and patient-friendly treatment and prophylaxis of AD and likely of psoriasis vulgaris as well.

## 4. Addressed Therapeutic Targets in Keratinocytes

1. Biosynthesis of Ceramides

The first therapeutic target is the biosynthesis of ceramides. The *de novo* synthesis of individual ceramides is performed by ceramide synthases 1–6 (CerS1–CerS6) at the endoplasmic reticulum. The type of synthesized ceramide(s) depends primarily on the fatty acid specificity (acyl-CoA specificity) of the CerSs that are present, the availability of different acyl-CoAs for incorporation as fatty acid side chain, and palmitoyl-CoA for the assembly of the sphingosine backbone [26,27]. Notably, a successful therapeutic approach has to ensure the availability of the required very long-chain acyl-CoAs (C_24_ to C_30_) (Figure 2). Keratinocytes in the epidermis of mice predominantly contain CerS3 (24%) and CerS4 (55.4%); CerS3 converts C_18_-CoA to C_32_-CoA and CerS4 C_18_-CoA to C_22_-CoA. Furthermore, according to recent studies, C_24:0_-CoA and C_24:1_-CoA are being incorporated [26,28]. Due to a lack of specificity for palmitoyl-CoA, both CerS3 and CerS4, which are predominantly present in skin, are not the reasonable point of origin of the pronounced increase of C_16_-ceramide in AD patient skin.

Ceramides with phytosphingosine backbone are formed by sphingolipid-C4-monooxygenase (hDES2) from ceramides with sphingosine backbone [29]. Phytosphingosine ceramide composition is closely related to ceramide *de novo* synthesis. Therefore, the normalization of sphingosine ceramide biosynthesis results in the normalization of phytosphingosine ceramides as well.

2. Composition of Free Fatty Acids

The second therapeutic target is closely linked to ceramide *de novo* synthesis and addresses alterations in the composition of free fatty acids due to impairment in very long-chain fatty acid biosynthesis. Very long–chain fatty acids (≥C_24_) are substantially reduced, while shorter fatty acids, in particular palmitic acid (C_16:0_) and stearic acid (C_18:0_), are increased [15]. Both fatty acids are preferably released by palmitoyl-CoA hydrolase (long-chain fatty-acyl-CoA hydrolase), unless their corresponding acyl-CoAs are processed by very long-chain-3-oxoacyl-CoA synthases (ELOVL1–ELOVL7) to obtain more elongated acyl-CoAs. Of the well-known ELOVLs, ELOVL7 (preferably C_16_–C_22_ acyl-CoA), ELOVL3 (especially C_18_ acyl-CoA) and ELOVL6 (chain extension C_16_-CoA to C_18_-CoA) are of particular interest. Although NADPH is not a cofactor of ELOVLs, NADPH activates the enzymatic activity of ELOVL6 and ELOV7 up to 10-fold. 3-ketoacyl-CoA reductase is responsible for the downstream reduction step in which NADPH is an enzyme cofactor, and activates ELOVL6 threefold [30]. Therefore, the abundant availability cytosolic NADPH is important for ELOVL elongation activity and the assembly of very long-chain ceramides. In cells lacking NADPH, C_16_-CoA is not converted to acyl-CoA ≥ C_18_ and is accessible to other reactions such as hydrolysis by palmitoyl-CoA hydrolase (Figure 4).

3. Availability of NADPH

Ensuring the availability of reduction equivalents (NAD(P)H) is the third therapeutic target, which is closely linked to the composition of free fatty acids (second target). NADPH is a powerful cellular reductant that is linked via mitochondrial nicotinamide nucleotide transhydrogenase (NNT) with NAD^+^, which is a mitochondrial proton acceptor and oxidant. NNT connects the hydride transfer between NADH and NADP^+^ (to NADPH and NAD^+^) across the inner mitochondrial membrane. Under physiological conditions, energy originates from the mitochondrial proton gradient. NAD^+^ is more than just an electron acceptor in mitochondrial energy generation. NAD^+^ acts as a substrate in mono(ADP)-ribosylation and poly(ADP)-ribosylation reactions of proteins and histones in the nucleus of cells as well. Ribosylation reactions are involved in cell regulation and the repair of single-strand breaks of DNA. Additional metabolites, including nicotinamide, free mono(ADP)-ribose, and poly(ADP)-ribose, can be derived from NAD^+^ [31].

ATP synthesis in mitochondria, glutathione (GSH) regeneration, and intracellular redox potential are directly linked to the availability of NAD^+^ and NADH/NAD^+^ levels. The involvement of NAD^+^ in these regulatory processes requires a constant re-synthesis of NAD^+^ to avoid exhaustion of the intracellular NAD^+^ pool. The availability of sufficient GSH is in turn necessary to neutralize reactive oxygen species (ROS) or reactive carbonyl species (RCS) to prevent DNA single-strand breaks.

4. Specific Generation of C_16_-ceramide

The fourth therapeutic target is the root cause of the specific generation of C_16_-ceramide in the skin of patients with AD and apoptotic cells. Unlike ceramides C_18_ to C_24_, it is not formed by *de novo* synthesis at the endoplasmic reticulum. CerS3 and CerS4 present in skin exhibit no selectivity for palmitoyl-CoA, which is required for the synthesis of C_16_-Ceramide [26]. The origin of C_16_-ceramide is located in the lysosome. Under physiological conditions, sphingomyelin from the cell membrane is degraded in the lysosome by acidic sphingomyelinase (aSMase) to phosphocholine and ceramide. In turn, the resulting ceramide is directly hydrolyzed by aCERase to sphingosine and free fatty acid. In AD patients, predominant sphingomyelins 16:0-SM and 24:1-SM are increased [21], indicating the inhibition of degradation by lysosomal enzymes aSMase and aCERase. Membranous sphingomyelin is ruled out as being of (C_16_-) ceramide origin.

## 5. Therapy Objectives for Treatment and Prophylaxis

Critical pro-apoptotic C_16_-ceramide in the skin of AD patients originates from a different cellular compartment than the (very) long-chain ceramides (>C_20_), which are responsible for establishing the dermal lipid barrier. The objectives of successful active adjustment of the ceramide metabolism are:Inhibition of forming C_16_-ceramide in lysosomes.Stabilization of very long-chain fatty acid-CoA synthesis (≥C_18_) for very long-chain ceramide *de novo* synthesis.Protecting lysosomal V-ATPase from inactivation (formation of the disulfide bond between Cys 254 and Cys 532) and stabilizing the ceramidase activity of aCERase.Modulating expression of AD-relevant gene.Enrichment of very long-chain ceramides of *de novo* synthesis.

## 6. Implementation of Therapy Objectives

1. Inhibition of forming C_16_-ceramide in lysosomes.

The synthesis of C_16_-ceramide in lysosomes can be blocked by lysosomotropic compounds. Well-known antidepressants imipramine, desipramine, and amitriptyline, together with the carbazole NB06 belong to this class of compounds (Figure 5) [32]. Lysosomotropic compounds are characterized by raising the lysosomal pH above 6 or higher. Thus, lysosomal enzymes are inactivated, and the lysosomal substrate supply of revaCERase is capped. Lysosomotropy is a physico-chemical property of a compound and is independent of a specific therapeutic indication. Inhibition of the formation of C_16_-ceramide likewise prevents premature apoptosis [11,32,33,34]. An overview of lysosomotropic compounds and their field of indication can be found in [35,36].

Lysosomotropy leads to an inhibition of the aCERase hydrolase activity by increasing the pH value. Very long-chain ceramides from *de novo* synthesis will no longer be degraded in the lysosome and will accumulate [33]. Although the lysosomal pH value is within the optimum range of revaCERase activity, only small amounts of C_16_-ceramide are formed due to a lack of substrate palmitic acid.

2.Stabilization of very long-chain fatty acid-CoA synthesis (≥C_18_) for very long-chain ceramide *de novo* synthesis and3.Protecting lysosomal V-ATPase from inactivation and stabilizing the ceramidase activity of aCERase

Stabilization of the synthesis of very long-chain ceramides and the protection of lysosomal V-ATPase as well as the stabilization of aCERase function are closely linked via NAD^+^ and NADPH. It is important to stabilize or reduce cytosolic redox potential to, or to below, a level that protects both cysteines of V-ATPase from oxidation and ensures sufficient amounts of NADPH and ATP for the individual reduction steps of the *de novo* synthesis of very long-chain ceramides and the lysosomal proton pumps.

One of the topical active compounds that can be used to implement the therapeutic concept is linoleic acid (Figure 6). Linoleic acid can serve as a radical scavenger or is available via linoleate ∆12-cis-∆11-trans-isomerase for the *β*-oxidation of fatty acids to recover reduction equivalents (NADH/NADPH) and/or energy (ATP). In general, the use of gamma-linolenic acid (evening primrose seed oil) is also feasible, but nerve-sensitizing prostaglandins (e.g., PGD2 and PGE2) can easily be formed via arachidonic acid, thus increasing itching. The maintenance of lysosomal pH ensures the stability of the ceramide hydrolase functionality of aCERase. No C_16_-ceramide that induces apoptosis can be formed.

4. Modulating expression of AD-relevant genes

Another important objective is the modulation of cytokines and interleukins involved in the development of AD. Anti-apoptotic lysosomotropic compounds not only interfere with classical lysosomal metabolic processes such as proteolysis and degradation of membrane lipids, but also have a modulating effect on the release or formation (transcription at gene level or maturation) of cytokines and interleukins such as CXCL3, CXCL2, CCL4, CCL20, CXCL10, PTX3, PTGS2, ICAM1, IL23A, IL6, and TNFalpha, which are involved in a variety of biological processes (apoptosis, lipid metabolism, cytokine secretion, cell activation, cell death, cell adhesion, cell differentiation, cell stimulation, and cell proliferation) [32]. In contrast to antibodies, small lysosomotropic chemical compounds such as NB 06 or amitriptyline can be used as local therapeutics for inflammatory skin or lesions. Thus, the self-perpetuating inflammatory process in keratinocytes can be interrupted.

5. Enrichment of very long-chain ceramides of *de novo* synthesis

A spin-off effect of lysosomotropic compounds is the accumulation of very long-chain ceramides by the inhibition of the ceramidase activity of aCERase [32]. However, this requires the formation of very long-chain ceramide in ceramide *de novo* synthesis. Obviously, the enrichment is closely linked to a stable, or with reduction equivalents stabilized, very long-chain ceramide *de novo* synthesis. Without an adequate supply of redox equivalents by linoleic acid, an enrichment using the lysosomotropic compounds amitriptyline or NB 06 is meaningless. C_16_-CoA, being unable to be converted through the lack of NADPH through ELOVL6 and ELOVL7, is hydrolyzed to palmitic acid and thus together with sphingosine is available in the lysosome for C_16_-ceramide synthesis. Due to the lysosomotropic compound, the pH in the lysosome is elevated to 6, which is in the optimum range of activity of the revaCERase (ceramide synthase activity). Now, there is no lack of substrates, and C_16_-ceramide can be formed.

While the preliminary stages of AD may be treated effectively with mere lysosomotropic compounds, the prevention of C_16_-ceramide increase in keratinocytes using lysosomotropic compounds alone is insufficient, and even risky, in moderate to severe forms and acute exacerbations, as they may provoke severe but reversible adverse effects (acute generalized exanthematous pustulosis) [37]. Sertraline is a lysosomotropic compound that is similar to amitriptyline [36], which exhibits lysosomotropic properties in keratinocytes if orally administered (50 mg per day). Therefore, it is obvious that the lysosomotropy of sertraline plays an important role in the development of the severe adverse effects that have been reported. A lack of redox equivalents NADPH and GSH in combination with pharmacologically-induced lysosomotropy is linked to the severe adverse effect that is observed. This implies that only the combination of a recovery of reduction equivalents (linoleic acid) and lysosomotropic active compound (amitriptyline) can result in an efficient accumulation of very long-chain ceramides in affected skin.

## 7. Therapeutic Concept

The customizable therapeutic concept consists of two modules: therapy and prophylaxis. The therapy combines two components: 1. lysosome protection and stabilization of very long-chain ceramide *de novo* synthesis, and 2. lysosomotropic and anti-apoptotic treatment of keratinocytes. Depending on the severity, treatment consists of a combination of components 1 and 2 (lesions, mild to moderate AD) or merely component 1 (initial redness, (mild) itching, feeling of tension). Prophylaxis is typically performed with component 1. Prophylaxis, on the other hand, has no time limit, and is also useful and recommended even after symptoms have subsided. Exclusive treatment with component 2 is insufficient and may cause itching and pustules, as previously described in Section 6.

On the basis of the suggested foundations listed in the European guidelines for treatment of AD, a light, moisturizing oil-in-water emulsion without the addition of urea and glycerin is used as the foundation of the cream. Fatty or occlusion-promoting foundations were excluded, since a poor release of active compounds is to be expected.

Therapeutic targets, the implementation of therapy objectives, and the active compound(s) applied for each purpose are summarized in Figure 7.

## 8. Personalized Therapy

The therapeutic concept presented can be easily customized in its composition and adapted to the individual requirements of patients. Variations in the composition of the cream can be: the concentration of linoleic acid, the concentration of the lysosomotropic compound, and its combination and the lysosomotropic compound itself, if additional pharmacological effects are advantageous. The suitable choice of foundation can be guided by the requirements of the individual patient, as is currently common with individual formulations. However, the basis should not be too rich (oily). Starting from a standard formulation, adjustments can easily be made for individual prescriptions. Semi-solid preparations are easy to prepare in pharmacies.

The verification of therapy success (response to therapy) can be performed by comparing the ceramide profiles of samples obtained from intact skin, reddened skin, and lesions in addition to the usual scores (e.g., EASI-score). Alternatively, a gene expression profile of specific genes (e.g., PTG2, CXCL10, IL23A, or IL8) is also appropriate.

A crucial factor for the duration of treatment is the time frame of approximately 28 days that the skin requires for complete regeneration. In more severe or chronic cases, the therapeutic effect may only occur after a half or full regeneration cycle of the epidermis (due to the therapeutic target skin).

## 9. Does the Concept Work?

Dissatisfaction with therapy, frustration, and speculation about the root causes of the disease are well-known problems in the treatment of AD patients. Many therapeutic approaches address the disease symptomatically, including inflammation-glucocorticoids, itching-H_1_ receptor antagonists or local anesthetics (polidocanol), and fail to alleviate painful itching and restore skin integrity. A typical example is mild AD and chronic AD on the crook of the arm and wrists with spontaneous worsening. Various topical therapy recommendations following the European guidelines for treating AD have been tried (emollient therapy, basic therapy, and glucocorticoids), but severely itching areas on the crook of the arm and wrists have not improved over three years. However, neither the restoration of lesional skin nor the full recovery of skin integrity was achieved with the European guidelines for treating AD. This was the major reason to validate the ceramide metabolism rested therapy concept in an off-label use [38]. With a combination of amitriptyline (lysosomotropic and anti-apoptotic) and linoleic acid (lysosome protection and very long-chain ceramides *de novo* synthesis stabilization) in therapy and linoleic acid in prophylaxis, the lesions vanished completely after approximately 60 days (two regeneration cycles of the skin), and have not recurred since then due to the frequent application of the prophylactic cream containing linoleic acid.

## 10. Conclusions

Despite new target-oriented anti-inflammatory IL4RA, IL13, IL31RA and IL23 antibodies [39] for severe cases, a holistic concept for the topical treatment of AD and regeneration of the skin’s barrier function at the cellular level is still lacking. With our paper, we wish to initiate a discussion about this topic and break new ground in the therapy and prophylaxis of AD and psoriasis. The possibility of a personalized therapy and the individual manufacture of creams provide new therapeutic tools that are beneficial to patients.

The modular and personalized therapy concept described can be used as a valuable complementary treatment option to restore the skin, strengthen the lipid skin barrier, and prevent dry skin as a precursor of AD. It is well-suited for patients with local lesions and independent of acute or chronic infections. A systemic effect by one of the active compounds and the detection in blood is unlikely. The combination of linoleic acid with amitriptyline offers the opportunity to treat dry and itchy skin as well as mild to moderate AD without the serious side effects of corticosteroids and prevent recurrence. Therefore, the described therapeutic approach is applicable for all age groups.

## Figures and Tables

**Figure 1 ijms-20-03967-f001:**
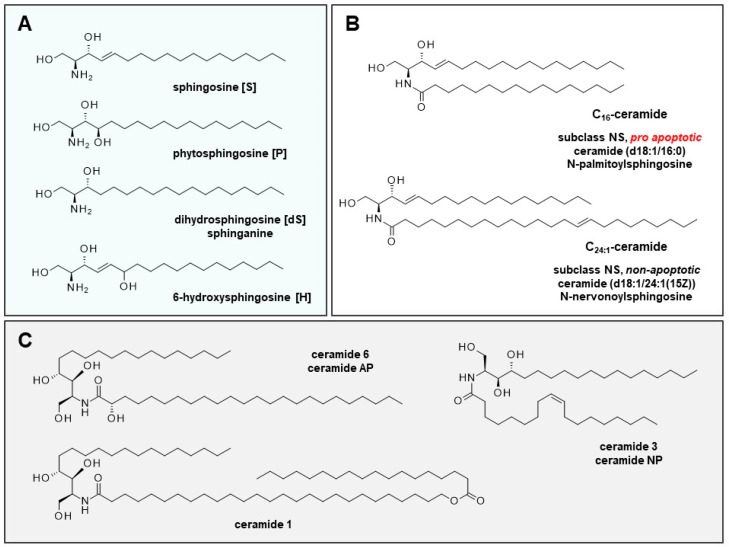
Backbones of dermal ceramides, cell cycle-relevant ceramides, and ceramides in basic ceramide replacement therapy. (**A**) Ceramide fractions of the stratum corneum are classified into 12 subclasses. The identifier is made up of the backbone (sphingosine [S], phytosphingosine [P], or 6-hydroxy-sphingosine [H], and the fatty acid residue (non-hydroxy fatty acid [N], α-hydroxy fatty acid [A] or esterified ω-hydroxy fatty acid [EO]). (**B**) Cell cycle-relevant ceramides C_16_ and C_24:1_ belong to the subclass NS (7% of dermal ceramide). Subclasses NH (23%), NP (22%), AP (16%), and AH (15%) are the most prominent ceramides in the stratum corneum. In AD, the AP subclass is significantly increased; in psoriasis, the AP, NP, and EOP subclasses are significantly lowered [6]. (**C**) Various creams for the treatment of affected skin areas comprise ceramide 1 (EOP) (N-(27-stearoyloxy-heptacosanoyl)-phytosphingosine), ceramide 3 (NP) (ceramide NP, N-oleoylphytosphingosine), and ceramide 6 (AP) (ceramide AP, N-(2-hydroxytetracosanoyl)-phytosphingosine for substitution therapy.

**Figure 2 ijms-20-03967-f002:**
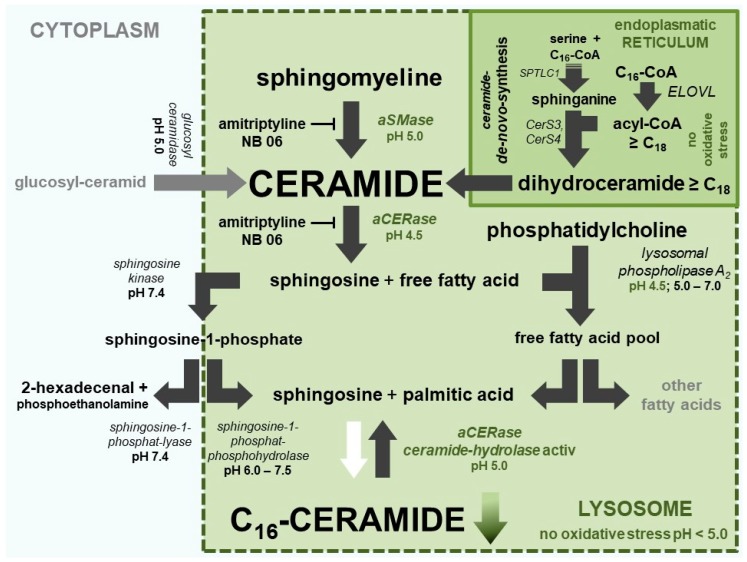
Ceramide metabolism in keratinocytes. Active pathways are marked with black arrows and inactive pathways are marked with white arrows. The objective of the ceramide metabolism-centered therapeutic concept is to use appropriate compounds to maintain lysosomal NADH-dependent redox chain, V-ATPase activity, and very long-chain ceramide *de novo* synthesis at the endoplasmic reticulum to prevent formation of pro apoptotic C_16_-ceramide in lysosomes.

**Figure 3 ijms-20-03967-f003:**
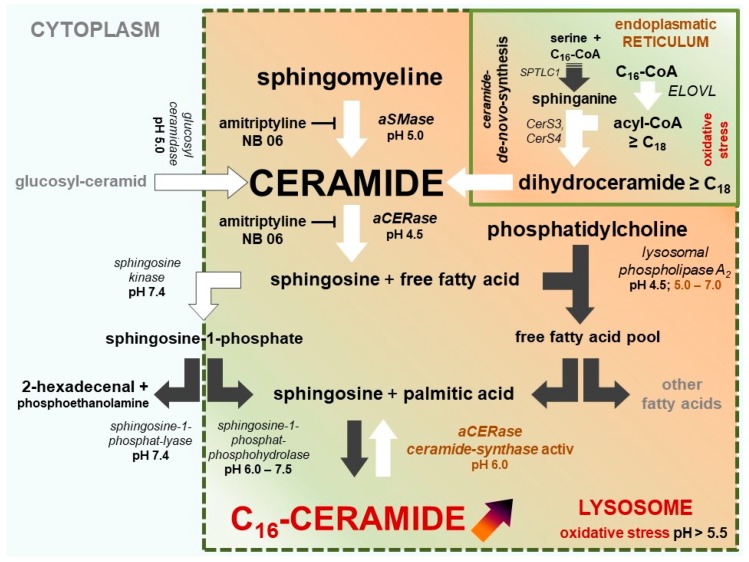
Ceramide metabolism and selective C_16_-ceramide synthesis in keratinocytes in atopic dermatitis lesions and sensitive/inflammatory skin (compartmentalized model of ceramide synthesis). Active pathways are marked with black arrows, less active and inactive pathways are marked with white arrows. Oxidative stress in keratinocytes results in the dysfunction of lysosomal V-ATPase. Ceramide hydrolase activity of acidic ceramidase (aCERase) loses activity and becomes inactive. In contrast, C_16_-ceramide is selectively re-synthesized by the reverse ceramide synthase activity of aCERase and accumulated in the lysosome. At the same time, very long-chain ceramide *de novo* synthesis is blocked due to insufficient NADPH. NADPH is an essential cofactor for very long-chain-3-oxoacyl-CoA synthases (ELOVL) for the synthesis of very long-chain acyl-CoAs (≥C_18_).

**Figure 4 ijms-20-03967-f004:**
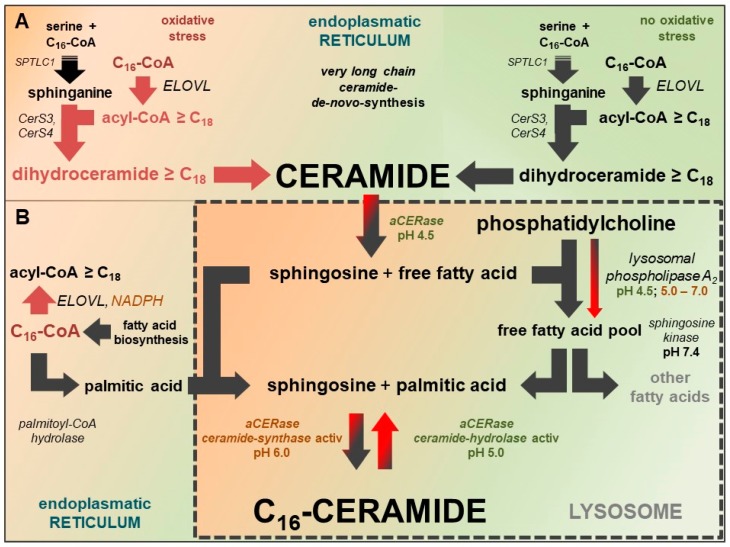
Linkage between synthesis of very long-chain fatty acids and very long-chain ceramides at the endoplasmatic reticulum and substrate supply of reverse ceramide synthase activity of aCERase for selective C_16_-ceramide synthesis in lysosomes. Active pathways at standard conditions are marked with black arrows, inactive pathways are marked with red arrows. Black and red arrows indicate pathways with diminished activity. (**A**) Very long-chain fatty acid synthesis at standard conditions (highlighted in green) and a lack of NADPH (oxidative stress) (highlighted in red). In cells lacking NADPH, C_16_-CoA is not converted to acyl-CoA ≥ C_18_ and accessible to other reactions. (**B**) The potential sources of palmitic acid for the selective lysosomal synthesis of C_16_-ceramide by reverse ceramide synthase activity of aCERase are: hydrolysis of accumulating C_16_-CoA (palmitoyl-CoA hydrolase), hydrolysis of phosphatidylcholine (lysosomal phospholipase A_2_), and sphingomyelin of cell membranes. Lysosomal phospholipase A_2_ has residual activity, even at pH values up to 7.0.

**Figure 5 ijms-20-03967-f005:**
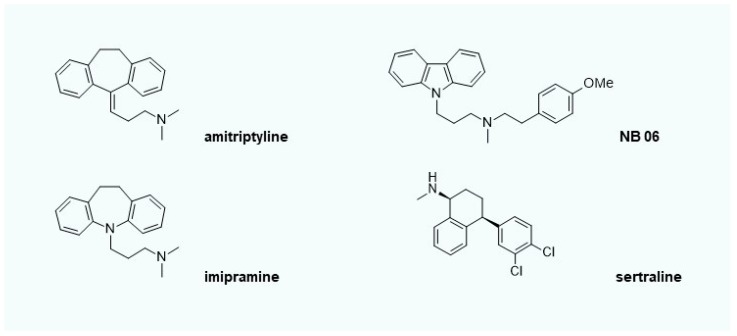
Lysosomotropic compounds amitriptyline, imipramine, sertraline, and the model compound NB 06 [32,36].

**Figure 6 ijms-20-03967-f006:**
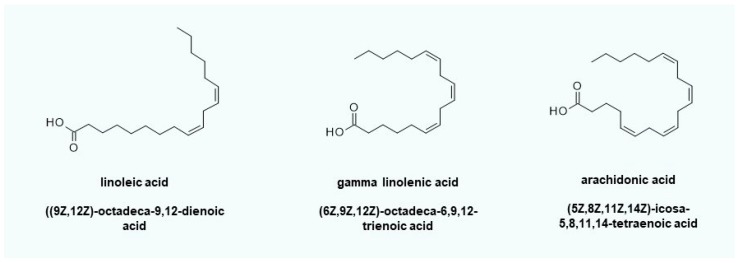
Fatty acids to protect the lysosomes and stabilize very long-chain ceramide *de novo* synthesis. Gamma-linolenic acid can be converted into dihomo-gamma-linolenic acid by elongation and subsequent oxidation to arachidonic acid by acyl lipid (8–3) desaturase. Arachidonic acid itself is a substrate of COX2 (PTGS2), which is inducible by oxidative stress. The resulting prostaglandin PGH2 and its derivatives can sensitize neurons. Linoleic acid first has to be converted into gamma linolenic acid by linoleoyl-CoA desaturase before it can be metabolized toward arachidonic acid. This can be a key advantage if itching or prostaglandin-mediated inflammatory reactions occur due to the application of creams containing gamma-linolenic acid.

**Figure 7 ijms-20-03967-f007:**
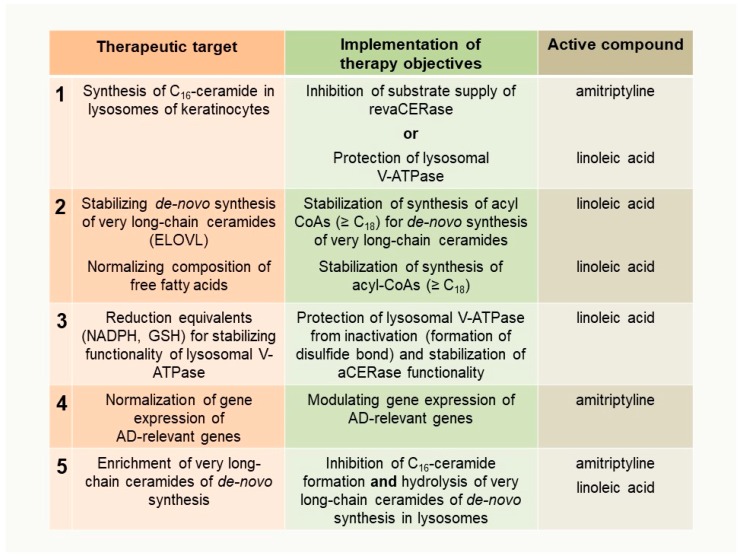
Derived therapeutic targets and the implementation of therapy objectives to reduce keratinocytes with disturbed maturation and premature apoptosis in the stratum corneum of AD patients. Therapeutic targets are listed in chronological order in Section 5 and Section 6.

**Table 1 ijms-20-03967-t001:** Most significant anomalies in the (sphingo)lipid profile of inflammatory skin and in lesions of atopic dermatitis patients.

Ceramides	Ceramides + Free Fatty Acids	Free Fatty Acids
Short chain ceramides, in particular C_16_-ceramide [NS], are elevated in inflammatory skin and lesions	Free fatty acids and ceramides with very long chains (≥C_24_) are significantly reduced.	Short-chain free fatty acids palmitic acid (C_16:0_) and stearic acid (C_18: 0_) are elevated.

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
