# Peer review of "Derailed Ceramide Metabolism in Atopic Dermatitis (AD): A Causal Starting Point for a Personalized (Basic) Therapy"

_ijms, 2019, doi:10.3390/ijms20163967_

Round 1

Reviewer 1 Report

Exact terminology is "rosacea" and "acute generalized exanthematous pustulosis".

Author Response

Response to Reviewer 1 Comments:

Response: We carefully checked the manuscript for misleading wording, terminology and edited it.

Reviewer 2 Report

This is an interesting review of  the role of ceramides in AD, and presents a case for using lysosomotropic agents as well as linoleic acid as a therapy for AD.

The paper is rather difficult to read, not because the English is bad, but rather because the abbreviations for the numerous enzymes and intermediates in ceramide synthesis required the reader to keep going back to previous graphs in order to follow along. This isn't really a criticism of the work; just the reality of following a complex pathway.

The authors cite the use of their proposed therapeutic approach on one patient, and have indicated that the results from this case report are being published elsewhere. It would have been interesting to have more details about the topical formulation that was used on this patient. That is, what was the concentration of amitriptyline and linoleic acid used on the patient and how often was it applied.

A few sentences were not very clear because of the choice of English words used, and this led to a slower reading of the paper. For example, in the introduction, the sentence "Morphology and localization is age-dependent pronounced differently and usually accompanied with severe itching". I had no idea what this sentence meant, and I assume it was the word "pronounced" that led to the confusion. Similarly, in section 3.1.1. the sentence "Furthermore, according to recent studies C24.0 and C24.1-CoA are being implemented". That sentence was also not clear.

The authors suggest using linoleic acid partly as a free radical trap. Given this, is there any evidence that topical formulations high in antioxidants are beneficial for treating AD. Or is the primary role of linoleic acid as a source of long chain fatty acids. 

Author Response

Response to Reviewer 2 Comments

The authors cite the use of their proposed therapeutic approach on one patient, and have indicated that the results from this case report are being published elsewhere. It would have been interesting to have more details about the topical formulation that was used on this patient. That is, what was the concentration of amitriptyline and linoleic acid used on the patient and how often was it applied.

Response: The case report will be published in September 2019. Pharm. 2019, 74, DOI: 10.1691/ph.2019.9484. The final PDF is not available yet. Applied concentrations were as follows: amitriptyline 0.03%, linoleic acid 0.5%, in base cream DAC (O/W cream), twice a day thinly. Other o/w creams are possible.  

A few sentences were not very clear because of the choice of English words used, and this led to a slower reading of the paper. For example, in the introduction, the sentence "Morphology and localization is age-dependent pronounced differently and usually accompanied with severe itching". I had no idea what this sentence meant, and I assume it was the word "pronounced" that led to the confusion. Similarly, in section 3.1.1. the sentence "Furthermore, according to recent studies C24.0 and C24.1-CoA are being implemented". That sentence was also not clear.

Response: We have checked and edited the manuscript for terminology and misleading wording.

The authors suggest using linoleic acid partly as a free radical trap. Given this, is there any evidence that topical formulations high in antioxidants are beneficial for treating AD. Or is the primary role of linoleic acid as a source of long chain fatty acids.

Response: Commercially available care products contain tocopherol(acetate), BHT or Licochalcon A as radical scavengers. We have tried to stabilize the more severely affected wrist of the publication with these products. Without success. Evening primrose seed oil (gamma linolenic acid) also failed to achieve the desired stability of the skin. Using 0.5% linoleic acid cream, the skin was stable for extended periods. As soon as stress situations occured, the wrist started itching again. Best results were achieved with linoleic acid. Therefore, both factors are important: substrate of beta-oxidation of fatty acids and radical scavenger.

For further information please feel free to contact us. Cooperation partners are welcome.